

# Polarity and epithelial-mesenchymal transition of retinal pigment epithelial cells in proliferative vitreoretinopathy

Hui Zou, Chenli Shan, Linlin Ma, Jia Liu, Ning Yang and Jinsong Zhao

Eye Center, The Second Hospital of Jilin University, Changchun, China

## ABSTRACT

Under physiological conditions, retinal pigment epithelium (RPE) is a cellular monolayer composed of mitotically quiescent cells. Tight junctions and adherens junctions maintain the polarity of RPE cells, and are required for cellular functions. In proliferative vitreoretinopathy (PVR), upon retinal tear, RPE cells lose cell-cell contact, undergo epithelial-mesenchymal transition (EMT), and ultimately transform into myofibroblasts, leading to the formation of fibrocellular membranes on both surfaces of the detached retina and on the posterior hyaloids, which causes tractional retinal detachment. In PVR, RPE cells are crucial contributors, and multiple signaling pathways, including the SMAD-dependent pathway, Rho pathway, MAPK pathways, Jagged/Notch pathway, and the Wnt/β-catenin pathway are activated. These pathways mediate the EMT of RPE cells, which play a key role in the pathogenesis of PVR. This review summarizes the current body of knowledge on the polarized phenotype of RPE, the role of cell-cell contact, and the molecular mechanisms underlying the RPE EMT in PVR, emphasizing key insights into potential approaches to prevent PVR.

## INTRODUCTION

Proliferative vitreoretinopathy (PVR) is a complex blinding disease that occurs after rhegmatogenous retinal detachment (RRD), surgical interventions, or ocular trauma. As a prolonged and exaggerated scarring process, PVR is characterized by the formation of contractile fibrocellular membranes in the vitreous cavity and on the inner and outer surfaces of the retina (*The Retina Society Terminology Committee, 1983*; *Mudhar, 2020*; *Tosi et al., 2014*). At present, surgical interventions, including vitrectomy, membrane peeling, pneumatic retinopexy, and scleral buckle, remain the mainstay of treatment in PVR. Although work in recent decades has led to advancements in surgical techniques and management, PVR cannot be effectively treated and is still the most common cause of failure to reattach the retina (*Coffee, Jiang & Rahman, 2014*; *Khan, Brady & Kaiser, 2015*; *Mitry et al., 2012*; *Wickham et al., 2011*). In addition, in spite of successful anatomic reattachment, the visual function of such cases cannot be improved, due to the retinal damage resulting from the mechanical contraction of fibrous membranes. Therefore, in order to improve postoperative visual function and reduce the incidence of this serious

Corresponding author
Jinsong Zhao,
jinsongzhao2003@163.com

complication, it is particularly important to explore new prophylactic and therapeutic approaches based on a deeper understanding of the pathogenesis of PVR.

A growing body of evidence indicates that the mechanisms of PVR are orchestrated by multiple elements (*Idrees, Sridhar & Kuriyan, 2019*; *Jin et al., 2017*; *Pastor et al., 2016*), such as growth factors (*Charteris, 1998*; *Ni et al., 2020*; *Pennock et al., 2014*; *Wubben, Besirli & Zacks, 2016*), cytokines (*Bastiaans et al., 2018*; *Harada, Mitamura & Harada, 2006*; *Limb et al., 1991*), extracellular matrix proteins (*Feist et al, 2014*; *Miller et al., 2017*) and various cells (*Eastlake et al., 2016*; *Pennock et al., 2011*; *Shu & Lovicu, 2017*). According to the histopathology of PVR, the fibrocellular membrane of PVR is composed of excessive extracellular matrix (ECM) and multiple types of cells, and retinal pigment epithelial (RPE) cells have been indicated as the most consistently present and the most abundant (*Amarnani et al., 2017*; *Ding et al., 2017*; *Hiscott et al., 1989*; *Machemer & Laqua, 1975*), proving that the RPE cell plays a crucial role in PVR. Under physiological condition, the polarized RPE cell is non-proliferative by cell–cell contact. However, when the eye suffers from a retinal break or trauma, RPE cells are exposed to various growth factors and cytokines that are produced by activated immune cells, leading to the disruption of junctional complexes in RPE cells. Subsequently, activated RPE cells detach from Bruch's membrane, migrate through the defect of the retina, proliferate, and transform into myofibroblasts, forming fibrotic membranes (*Chen, Shao & Li, 2015*; *Morescalchi et al., 2013*; *Palma-Nicolás & López-Colomé, 2013*). In an analogous process to exaggerated wound healing response, these membranes can attach to the retina and contract, resulting in further retinal detachment and poor vision (*Chiba, 2014*; *Garweg, Tappeiner & Halberstadt, 2013*). It is noteworthy that due to the loss of cell–cell contact, RPE cells undergo epithelial-mesenchymal transition (EMT), which is pivotal in the development of PVR. During EMT, RPE cells transdifferentiate into mesenchymal cells that are characterized by increased motility, and enhanced ability to proliferate, resist apoptosis and produce extracellular matrix proteins, thus participating in PVR (*Tamiya & Kaplan, 2016*; *Zhang et al., 2018c*). These indicate that in-depth knowledge of EMT may provide insight into potential approaches to prevent PVR. Therefore, this review focuses on the polarized phenotype of RPE and molecular mechanisms of RPE cell EMT, discussing the role of RPE cells in PVR.

## SURVEY METHODOLOGY

We used the PubMed database to search available literature based on keywords including "proliferative vitreoretinopathy(PVR)" and "retinal pigment epithelial cell". To include more information on the polarity of RPE, we also searched articles about the structure and function of cell–cell junctions in RPE cells that explored the role of cell–cell contact in EMT.

### The polarized retinal pigment epithelial cell

The human RPE cell achieves terminal differentiation at four to six weeks of gestation and subsequently remains mitotically quiescent (*Lutty & McLeod, 2018*; *Stern & Temple, 2015*). The RPE, which is situated between the photoreceptors and the choroid, plays many complex roles indispensable to the health of the neural retina and the choroid. These
roles include recycling of components of the visual cycle, absorption of light to protect from photo-oxidative stress, production of essential growth factors, immunological regulation of the eye, phagocytosis of photoreceptor outer segments generated during daily photoreceptor renewal, and transportation across the blood retina barrier (BRB) (*Ferrington, Sinha & Kaarniranta, 2016*; *Fields et al., 2019*; *Mateos et al., 2014*; *Naylor et al., 2019*; *Strauss, 2005*; *Vigneswara et al., 2015*). In order to maintain these multiple functions, RPE cells display a highly specialized structural and functional polarity.

Similar to other epithelia, the RPE displays three characteristics of the epithelial phenotype: apical plasma membrane, junctional complexes, and basolateral domain. RPE cells display structural polarity, with apical microvilli and melanosomes, and basal microinfolds. The abundant melanin granules in RPE cells absorb stray light, a process that is essential for visual function (*Strauss, 2005*). In a polarized cell, the distributions of surface proteins on the apical and basal plasma membranes are different, contributing to the performance of cellular functions (*Khristov et al., 2018*). However, a highly polarized distribution of ion channels, transporters and receptors in RPE is different from that observed in conventional extraocular epithelia (*Lehmann et al., 2014*). For example, Na, K-ATPase (*Sonoda et al., 2009*) and monocarboxylate transporters (MCT) 1 (*Deora et al., 2005*) are localized to the apical aspect of RPE cells, while chloride transporter CFTR (*Maminishkis et al., 2006*) is basally located. On the apical plasma membrane, RPE cells phagocytize the photoreceptor outer segments, which are regulated by polarized receptors. *Bulloj et al. (2018)* found that binding of Semaphorin 4D (sema4D) to RPE apical receptor Plexin-B1 suppresses outer segment internalization, contributing to the maintenance of photoreceptor function and longevity. The RPE also transports fluid out of the subretinal space, and regulates bidirectional nutrient transport between the outer retina and the choroid, in a manner dependent on the polarized distribution of membrane channels and transporters (*Strauss, 2005*). The RPE basolaterally secretes extracellular matrix components and factors, which participate in ECM remodeling and maintain the outer BRB (oBRB) function (*Caceres & Rodriguez-Boulan, 2020*). Therefore, the polarized phenotype of the RPE is vital to both the oBRB and is the basis of the homeostasis of the outer retina (*Caceres & Rodriguez-Boulan, 2020*; *Lehmann et al., 2014*). The disruption of RPE polarity contributes to the development of several retinal diseases, such as PVR and age-related macular degeneration (AMD). A comprehensive understanding of the way in which this polarity is achieved may provide insights into the pathogenesis of PVR.

However, most available data on RPE polarity is contributed by studies performed on RPE-immortalized cell lines that show partial preservation of the RPE phenotype, and were extrapolated from data obtained from the prototype Madin-Darby Canine Kidney (MDCK) cell line (*Lehmann et al., 2014*). The detailed mechanisms that determine RPE polarization remain unclear. Some scholars believe that junctional complexes, including adherens junctions (AJs) and tight junctions (TJs), are essential for building epithelial cell polarity and maintaining the integrity of epithelial layers such as RPE (*Niessen, 2007*; *Pei et al., 2019*; *Tamiya & Kaplan, 2016*).

Tight junctions are complex cell–cell junctions formed by transmembrane proteins interactions with peripheral cytoplasmic proteins (Fig. 1). Transmembrane proteins

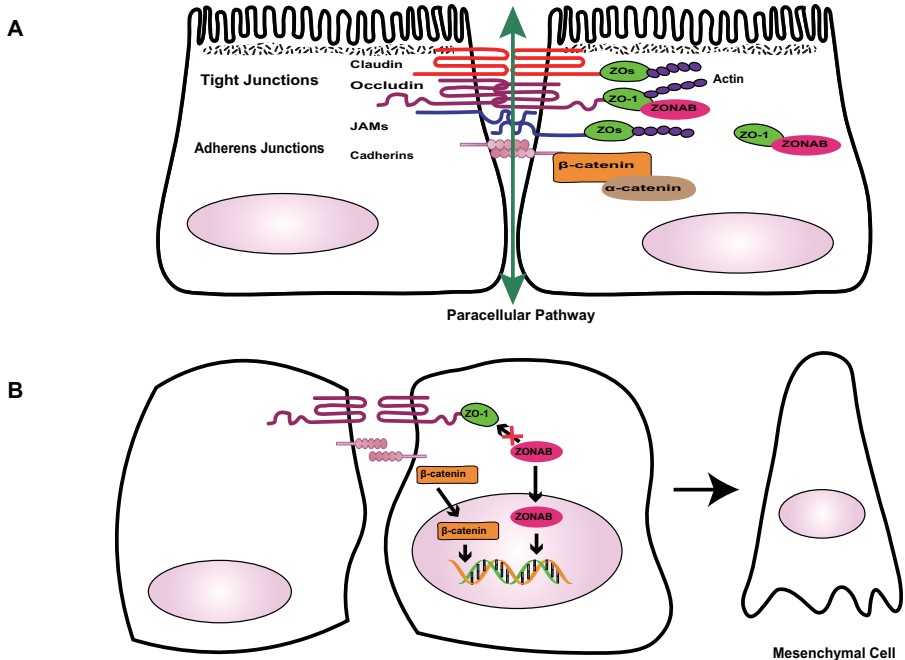

**Figure 1  Role of cell-cell contact in health and PVR.** (A) Tight junctions and adherens junctions maintain cell-cell contact and cell polarity in RPE cells. Mature RPE cells with cell-cell contact remain dormant by sequestering EMT effectors to prevent nuclear localization. ZO-1 sequesters nucleic acid-binding protein (ZONAB) at tight junctions/cytoplasm, and adherens junctions sequester β-catenin by binding to epithelial cadherins. Tight junctions have a barrier function that control the passage of solutes. (B) Loss of cell-cell contact initiates EMT. Deconstruction of junctional complexes or reduction of epithelial cadherins/ZO-1 elicits nuclear localization of ZONAB/β-catenin and activation of their target genes, and disrupts the outer blood retinal barrier, facilitating the release of growth factors and cytokines, which further aggravate PVR.

include occludin, members of the claudin family, and junctional adhesion molecules (JAMs). Peripheral cytoplasmic proteins, such as zonula occludens (ZOs), form bridges between transmembrane proteins and the actin filament cytoskeleton and play a key role in the assembly and organization of TJs (*Bazzoni & Dejana, 2004*; *Bazzoni et al., 2000*; *Naylor et al., 2019*).

The RPE tight junctions regulate the paracellular movement of solutes via size and charge selectivity (*Benedicto et al., 2017*; *Caceres et al., 2017*; *Naylor et al., 2019*).Occludin and claudins determine the permeability and semi-selectivity of the TJs, and as such play critical roles in the oBRB (*Balda et al., 2000*; *Fields et al., 2019*; *Furuse et al., 1998*; *Günzel & Yu, 2013*; *Rosenthal et al., 2017*). JAMs regulate TJ assembly and function by recruiting other proteins to the TJ and play an important role in the barrier property of TJs (*Balda & Matter, 2016*; *Orlova et al., 2006*; *Shin, Fogg & Margolis, 2006*). In patients with RRD, damage to TJs elicits the breakdown of oBRB and promotes the penetration of growth factors and cytokines, aggravating PVR. As well as having a barrier function, TJs define the physical separation between apical and basal domains of the plasma membrane, to maintain RPE cell polarity (*Campbell, Maiers & DeMali, 2017*; *González-Mariscal et al.,*

2014; *Sluysmans et al., 2017*). The two extracellular loops of occludin mediate adhesion of adjacent cells and block the movement of plasma components. The C-terminal domain combines directly with ZOs, subsequently interacting with the actin cytoskeleton, which is essential to organizing and maintaining cell polarization (*Balda & Matter, 2016*; *Furuse et al., 1994*; *Shin, Fogg & Margolis, 2006*; *Tarau et al., 2019*). *Feng et al. (2019)* demonstrated that during EMT, the breakdown of TJs resulting from loss of claudin-1 causes ARPE-19 cells to lose their epithelial phenotype and transform into fibroblasts, promoting the development of PVR. TJs are involved in the regulation of signaling pathways that govern various cellular functions such as proliferation, migration, and differentiation (*Bhat et al., 2018*; *Shi et al., 2018*; *Sluysmans et al., 2017*). *Vietor et al. (2001)* found that decreased amounts of occludin can cause up-regulation and translocation of the adhesion junction protein β-catenin, which interacts with the transcription factor lymphoid enhancer-binding factor (LEF)/T cell factor (TCF) in the nucleus, leading to a loss of the polarized epithelial phenotype in EpH4 cells. ZOs, adaptor proteins within the TJ complex, exhibit dual localization at TJs and in the nucleus. Under injury or stress, the disruption of TJs increases ZO-2 nuclear accumulation, driving its interaction with transcription factors, and inducing MDCK epithelial cell proliferation (*Islas et al., 2002*; *Shi et al., 2018*; *Traweger et al., 2003*). In differentiated RPE cells, the interaction between ZO-1 with ZO-1-associated nucleic acid-binding protein (ZONAB) maintains cell–cell contact by sequestering ZONAB at the TJ or in the cytoplasm, maintaining cells dormancy. However, when damage to TJs decreases ZO-1 levels, ZONAB is translocated into the nucleus, leading to the up-regulation of cyclin D1 (CD1) and subsequent cell proliferation (*Balda, Garrett & Matter, 2003*; *González-Mariscal et al., 2014*). Therefore, TJs provide a structural foundation for the maintenance of cell–cell contact. *Georgiadis et al. (2010)* demonstrated that the overexpression of ZONAB or knockdown of ZO-1 could result in increased RPE proliferation and the development of EMT. Recent research has confirmed that during EMT, ZO-1 is decreased in ARPE-19 cells, and the knockdown of either ZO-1 or AJ protein E-cadherin leads to the downregulation of the other protein, indicating the existence of an interaction between the two junctional complexes (*Bao et al., 2019*). Due to the importance of TJs in the maintenance of integrity and functionality of epithelial cells, several researchers have focused on novel factors that stimulate the formation of TJs, such as nicotinamide (*Hazim et al., 2019*) and lysophosphatidic acid (*Lidgerwood et al., 2018*). Studies into these factors may produce well-differentiated RPE cell lines and a platform to enable the rapid expansion of our understanding of many RPE functions and retinal pathologies. This approach could be conducive to finding novel therapeutic interventions for PVR.

Besides the TJ complex described above, another type of junctional complex called AJs plays a key role in the maintenance of the integrity of epithelial cells and cell–cell contact (Fig. 1). Cadherins, the major proteins of AJs, belong to the glycoprotein superfamily, of which there are more than 20 members. The cytoplasmic domain of cadherins regulates interactions between cadherins and catenins, including β-catenin, α-catenin, and p120-catenin, and other scaffolding proteins such as ZO-1, to maintain cell shape and modulate cell proliferation (*Aberle et al., 1994*; *Nelson & Nusse, 2004*; *Wheelock & Johnson, 2003*). In quiescent adult RPE cells, epithelial cadherins (E- and/or P-cadherin) sequester β-catenin

at the AJs to maintain cell–cell contact. Reduction of cadherin levels or dissociation of AJs allows β-catenin to translocate into the nucleus, where it interacts with the transcription factor LEF, and activates the transcription of various genes, including Snail and cyclin D1, which participate in RPE cell EMT via the canonical Wnt/β-catenin signaling pathway (*Gonzalez & Medici, 2014*; *Lamouille, Xu & Derynck, 2014*; *Nelson & Nusse, 2004*; *Yang et al., 2018*) . *Tamiya, Liu & Kaplan (2010)* suggested that the loss of P-cadherin causes the loss of cell–cell contact and initiates RPE cell migration and EMT. These events coincide with a switch in cadherin isoform expression from P- to N-cadherin. In addition, hepatocyte growth factor (HGF) and its receptor c-Met can destabilize cell–cell adhesion and elicit nuclear translocation of β-catenin, resulting in RPE cell migration (*Lilien & Balsamo, 2005*; *Liou et al., 2002*). Jin et al. found that HGF induces loss or redistribution of junctional proteins ZO-1, occludin, and β-catenin in RPE explants, potentially damaging barrier function and increasing the migration of RPE cells, resulting in retinal detachment(RD) and PVR (*Jin et al., 2002*; *Jin et al., 2004*). Given the importance of HGF in the interruption of RPE junction, HGF may be a potential target for the prevention and treatment of PVR. However, this possibility needs further study.

Under physiological conditions in the eye, TJs and AJs maintain the specialized structural and functional polarity of RPE cells and play a pivotal role in the maintenance of cell–cell contact; they sequester EMT signaling effectors ZONAB and β-catenin at the junction or cytoplasm to prevent cells from responding to mitotic factors, causing cells to leave the cell-cycle (Fig. 1). Thus, normally, RPE cells form a cobblestone-like monolayer of immotile, polarized, and mitotically quiescent cells. However, once junctional complexes break down, RPE cells undergo EMT, which is an important contributor to proliferative vitreoretinopathy. In this pathological process, RPE cells lose their structural and functional polarity and transdifferentiate into mesenchymal cells, which proliferate, resist apoptosis, possess migratory ability, and produce abundant ECM, leading to the formation of an aberrant scar-like fibrocellular membrane.

## De-differentiated RPE and fibrocellular membrane

Proliferative vitreoretinopathy is characterized by the formation of fibrocellular membranes composed of proliferative and migratory cells and excessive, aberrant ECM. Histopathological analysis of PVR has demonstrated that PVR membranes have contractile activity and strain the retina, leading to tractional retinal detachment (TRD), which is responsible for blurring vision.

Several studies (*Feist et al, 2014*; *Takahashi et al., 2010*) have found that the cellular components of PVR membranes include RPE cells, myofibroblasts, fibroblasts, glial cells and macrophages, and that myofibroblasts are critical for the formation and contractile activity of fibrocellular membranes. Based on the indirect immunofluorescence evaluation of human PVR membranes, *Feist et al (2014)* showed that myofibroblasts originate principally from RPE cells through EMT. Myofibroblasts are characterized by increased expression of alpha-smooth muscle actin (α-SMA) and incorporation of α-SMA into newly formed actin stress fibers, which enhances their contractile properties. Myofibroblasts also secrete excessive matrix and pro-fibrogenic factors, promoting the contraction of PVR

membranes that ultimately cause irreversible loss of vision (*Gamulescu et al., 2006*; *Hinz et al., 2001*; *Shu & Lovicu, 2017*; *Tamiya & Kaplan, 2016*; *Tomasek et al., 2002*).

In addition to myofibroblasts, abnormally increased ECM reinforces the continuous contractile tension of PVR membranes, and this mechanical tension, together with specialized ECM proteins, regulates myofibroblast differentiation and its function, contributing to PVR. In PVR membranes, the primary components of ECM are collagen and fibronectin. The majority of collagen fibrils are type I collagen, which is synthesized by RPE cells and Müller cells. Collagen fibrils provide tensile strength to the ECM, and activate Rho, resulting in the translocation of myocardin-related transcription factor (MRTF) into the nucleus and promoting RPE cell EMT (*Guettler et al., 2008*; *Miralles et al., 2003*). Fibronectin may also play a significant role in PVR. During pathological ECM remodeling, fibronectin is one of the earliest ECM components recruited, serving as a scaffold for other ECM proteins (*Kadler, Hill & Canty-Laird, 2008*; *Miller et al., 2017*; *Miller et al., 2014*). Extra domain (ED)-A fibronectin, a splice variant of fibronectin, is increased in transforming growth factor (TGF)-β2-induced RPE cells and induces myofibroblast differentiation, participating in PVR (*Khankan et al., 2011*).

Under normal conditions, ECM breakdown by proteases such as matrix-metalloproteases (MMPs) plays a crucial role in ECM remodeling and the release of growth factors, maintaining tissue homeostasis in cooperation with ECM synthesis, reassembly, and chemical modification (*Bonnans, Chou & Werb, 2014*; *Craig et al., 2015*; *Lindsey et al., 2016*). As mentioned above, the polarized RPE is able to basolaterally secrete the extracellular matrix components fibronectin and collagens, MMP and tissue inhibitors of MMPs (TIMPs), which participate in ECM remodeling. However, under pathological conditions such as inflammation and retinal injury, RPE cells lose their apical-basal polarity, undergo EMT and abnormally secrete MMPs, TIMPs and ECM proteins, leading to dysregulated ECM remodeling (*Greene et al., 2017*). Such ECM has aberrant composition and organization and mechanical properties, and enhances matrix stiffness and strain, which disrupts the normal structure and function of the retina, exacerbating the progression of PVR.

## RPE and epithelial-mesenchymal transition
### EMT of RPE cell

Epithelial-mesenchymal transition is an important biological process, in which epithelial cells transdifferentiate into mesenchymal cells. Although EMT can occur in normal embryonic development and wound healing, it also participates in pathological processes such as fibrosis, cancer progression, and PVR. There are three distinct subtypes of EMT: type 1 occurs during tissue and embryo development, type 2 is involved in wound healing and organ fibrosis, and type 3 is associated with cancer progression and metastasis (*Dongre & Weinberg, 2019*; *Kalluri & Weinberg, 2009*). This review focuses on type 2 EMT, which is crucial to PVR. During EMT, due to junctional complexes damage, RPE cells relinquish their apical-basal polarity, reorganize their cytoskeletal architecture, and convert into spindle-shaped cells (Fig. 1). These cells downregulate the expression of epithelial proteins such as E-cadherin and ZO-1, and increase expression of mesenchymal drivers including

N-cadherin, vimentin, α-SMA and fibronectin (*Li, Zhao & He, 2020*). This mesenchymal transdifferentiation of RPE cells can increase the directional motility of individual cells, confer resistance to apoptosis, and facilitate cell proliferation and dysregulated ECM remodeling, eventually leading to the formation of PVR membranes.

### Transcription factors of EMT

The details of the molecular mechanisms that drive RPE cell EMT and lead to PVR remain to be clarified. Emerging evidence suggests that diverse extracellular inductive signals, including soluble cytokines and growth factors, and ECM components, can modulate the expression and activity of EMT-associated transcription factors and act together to control the initiation and progression of EMT in responding epithelial cells (*Yang et al., 2020*). Among the various transcription factors involved in the induction of EMT, core transcription factors including Snail 1, Snail 2(also known as Slug), Twist 1 and zinc-finger E-box-binding (Zeb) 1 have been identified as important regulators of RPE cell EMT. These factors impact the expression of genes that control repression of the epithelial phenotype and activation of the mesenchymal phenotype (*Boles et al., 2020*; *Feng et al., 2019*; *Li et al., 2019*; *Li et al., 2014*; *Liu et al., 2009*; *Palma-Nicolás & López-Colomé, 2013*). For example, thrombin can repress the expression of E-cadherin by stimulating Snail 2 expression and promote the expression of N-cadherin by phosphoinositide 3-kinase (PI3K)/PKC-ζ/mTOR signaling in Rat RPE cells (*Palma-Nicolás & López-Colomé, 2013*). During RPE dedifferentiation in primary culture, Zeb1 is overexpressed and binds to the MITF A promoter to repress the cyclin dependent kinase inhibitor, p21CDKN1a, resulting in RPE cell proliferation and EMT (*Liu et al., 2009*). These EMT transcription factors often act in concert, functionally cooperating at target genes by the convergence of signaling pathways. However, the molecular details of how these transcription factors contribute to EMT are still elusive (*Lamouille, Xu & Derynck, 2014*; *Stone et al., 2016*).

### Epigenetic factors of EMT

Due to the importance of epigenetic regulation of EMT, epigenetic modifiers have attracted increasing attention. Evidence has shown that epigenetic modifiers work in concert with transcription factors at different molecular layers to regulate the EMT process (*Skrypek et al., 2017*). Several epigenetic factors have been described including DNA methylation, histone modification and non-coding RNA. Because of the specific machinery utilized for EMT activation, these modifications are characterized by cell type specificity. In RPE cells, Methyl-CpG-binding protein 2 (MeCP2), a DNA methylation reader, plays a crucial role in the induction of EMT, and DNA methylation may participate in the pathogenesis of PVR (*He et al., 2015*; *Li, Zhao & He, 2020*). *He et al. (2015)* found high levels of expression of MeCP2 in all human PVR membranes, and concluded that MeCP2 mediates α-SMA expression through Ras GTPase activating protein (RASAL1). Furthermore, DNA methylation inhibitor 5-Aza-2′ deoxycytidine (5-AZA-dC) reportedly inhibits the expression of TGF-β-induced α-SMA and FN in human fetal RPE cells. It appears that 5-AZA-dC may have therapeutic value in the treatment of PVR. However, the mechanisms underlying the blockade of α-SMA and FN expression are complex, and further investigation is warranted.

Recently, the role of histone modifications associated with EMT has been assessed in RPE cells. However, there has been little research into the regulation of RPE cell EMT by histone modification. *Boles et al. (2020)* reported that TGF-β1 and TNF-α co-treatment (TNT) induces an EMT program in adult human RPE stem cell (RPESC)-RPE cells, involving an apparent reorganization of H3K27ac and H3K4me1 patterns at distal enhancers. The regions that gain H3K27ac tend to have a high H3K4me1/H3K4me3 ratio, indicating that they have enhancer activity and are associated with upregulated genes. *Xiao et al. (2014)* found that the expression of histone deacetylases (HDACs) in TGF-β-induced EMT of RPE cells was increased, and that Trichostatin A (TSA), a class I and II HDAC inhibitor, attenuated TGF-β2-induced EMT by inhibiting the canonical SMAD pathway and the non-canonical signaling pathways, including Akt, p38MAPK, ERK1/2 pathways and Notch pathway. Therefore, histone modifications may participate in the regulation of RPE cell EMT, and HDAC inhibitors may have potential as drugs for the prevention and treatment of PVR.

The study of EMT mechanisms at the RNA level has provided new perspectives on the treatment of PVR (*Kaneko & Terasaki, 2017*; *Wang et al., 2016*). MicroRNAs (miRNAs) are small noncoding RNAs that contribute to cellular processes by regulating gene expression. In differentiated RPE cells, microRNA-204 is highly expressed, and represses the expression of type II TGF-β receptors and Snail 2, maintaining epithelial structure and function. In contrast, low expression levels of miR-204 and anti-miR-204 promote RPE cells proliferation, participating in EMT (*Wang et al., 2010*). MicroRNA-194 overexpression can also suppress RPE cell EMT by attenuating the expression of Zeb1 (*Cui et al., 2019*). In addition to miRNAs, long non-coding RNAs (lncRNAs) contribute to the regulation of RPE EMT (*Zhang et al., 2019*). In RPE cells treated with PVR vitreous or TGF-β1, MALAT1 expression is increased, and knockdown of MALAT1 attenuates the phosphorylation of SMAD2/3 and the expression of Snail, Slug, and Zeb1, preventing cell migration and proliferation (*Yang et al., 2016*). In patients with PVR, MALAT1 is increased in the blood, and is reduced after surgery. Thus, MALAT1 may be a potential prognostic and diagnostic indicator for PVR (*Zhou et al., 2015*).

### Signaling pathways of EMT

During RPE cell EMT, extracellular signals change the expression of genes encoding epithelial and mesenchymal proteins and mediate cellular behavior such as cell migration, proliferation, and apoptosis through a network of interacting signaling pathways that contribute to the development of PVR (*Chen et al., 2014a*; *Chen et al., 2014b*; *Lee-Rivera et al., 2015*). Among these, TGF-β and its intracellular cascades play a key role in the EMT of RPE cells.

TGF-β induces EMT of RPE cells via two pathways: the classical SMAD-dependent pathway and the SMAD-independent pathway (Fig. 2) (*Cai et al., 2018*; *He et al., 2017*; *Heffer et al., 2019*; *Ishikawa et al., 2015*; *Takahashi, Haga & Tanihara, 2015*; *Yao et al., 2019*; *Zhang et al., 2017*; *Zhang et al., 2018b*; *Zhou et al., 2017*). In the SMAD dependent pathway, TGF-β binds to cell surface receptor complexes, and activates type I TGF-β receptors, which phosphorylate SMAD2 and SMAD3. The activated SMADs combine

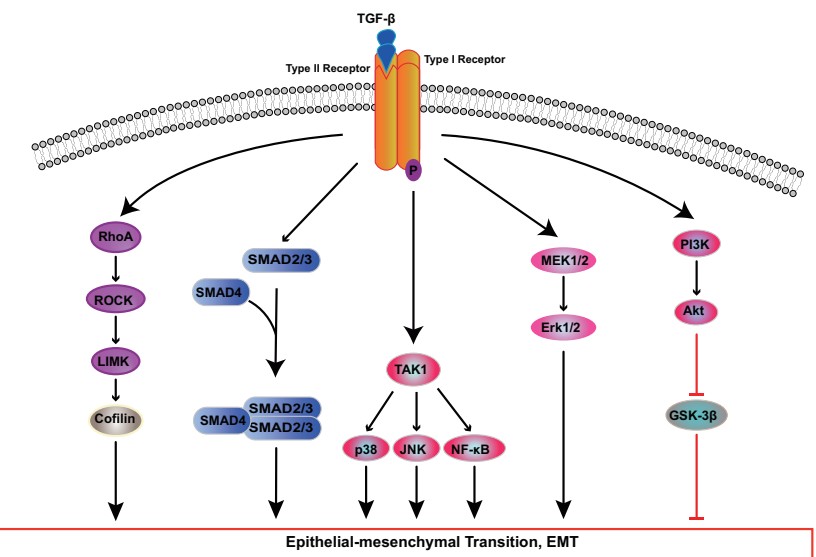

**Figure 2** **Signaling pathways of TGF-β-induced RPE cells EMT.** Transforming growth factor-β (TGF-β) activates various signaling pathways that cooperate to cause EMT. Besides canonical SMAD-dependent signaling, TGF-β can activate the Rho, PI3K/AKT, ERK MAPK, p38 MAPK, JUN N-terminal kinase (JNK) and nuclear factor-κB (NF-κB) pathways.

with SMAD4 to form a SMAD complex, which then enters the nucleus and combines with regulatory elements to regulate the expression of key genes associated with EMT. In addition to SMAD-dependent signaling, TGF β induces EMT through SMAD independent signaling pathways including Rho GTPase-dependent pathways (*Lee, Ko & Joo, 2008*), PI3K/Akt pathway (*Huang et al., 2017*; *Yokoyama et al., 2012*), mitogen-activated kinase (MAPK) pathways (*Chen et al., 2017*; *Lee et al., 2020*; *Matoba et al., 2017*; *Schiff et al., 2019*) and Jagged/Notch signaling pathway (*Zhang et al., 2017*). The MAPK signaling pathways include extracellular signal-regulated kinase(ERK) MAPK pathway, p38 MAPK pathway, and JUN N-terminal kinase (JNK) pathway (*Parrales et al., 2013*; *Schiff et al., 2019*; *Xiao et al., 2014*; *Zhang et al., 2018a*).

The Rho pathway has been reported to regulate the assembly and organization of the actin cytoskeleton and associated gene expression, and may be essential for the fibrotic response of RPE cells in PVR. In TGF-β1-treated ARPE-19 cells, activated RhoA or its downstream effector Rho kinase (ROCK) increase the kinase activity of LIM kinase (LIMK) which then phosphorylates cofilin. This phosphorylation attenuates the activity of cofilin, which promotes actin polymerization and reorganizes the actin cytoskeleton, leading to stress fiber formation (*Lee, Ko & Joo, 2008*). TGF-β-induced RhoA activation also facilitates cell migration and increases α-SMA expression in primary RPE cells (*Tsapara et al., 2010*). *Itoh et al. (2007)* demonstrated that ROCK inhibitor Y27632 and RhoA inhibitor, simvastatin, suppress TGF-β2-induced type I collagen expression in ARPE-19 cells, and confirmed the existence of crosstalk between the SMAD pathway and the Rho pathway. Some studies have suggested that activated SMAD3 induces NET1 gene expression to regulate RhoA activation in RPE cells (*Lee et al., 2010*). Moreover, thrombin can activate Rho and ROCK,

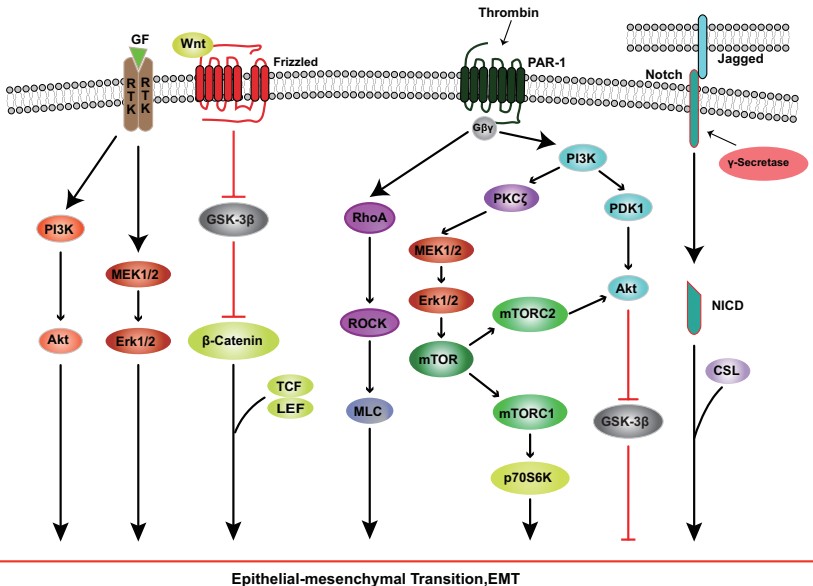

**Figure 3** **RTK, Wnt, Notch, and Thrombin signaling in RPE cells EMT.** Growth factors (GFs) stimulate receptor tyrosine kinases (RTKs) and induce EMT through PI3K-AKT and ERK MAPK signaling pathways. Thrombin activates PI3K and Rho signaling. PI3K promote EMT through Akt and mTOR pathways. WNT signaling promotes EMT by inhibiting glycogen synthase kinase-3β (GSK-3β) to result in nuclear localization of β-catenin, which interact with the transcription factors lymphoid enhancer factor (LEF) /T cell factor (TCF) and change genes expression. The intercellular interaction of Jagged ligands with Notch receptors induces EMT through the cleavage and release of the Notch ICD, which then activate target genes.

leading to myosin light chain (MLC) phosphorylation and actin stress fiber formation in EMT of RPE cells (Fig. 3) (*Ruiz-Loredo, López & López-Colomé, 2011*). Therefore, ROCK inhibitor and RhoA inhibitor may be new potential therapeutic target drugs for PVR.

The PI3K/Akt pathway mediates a broad range of cellular functions, such as cell transformation, migration, proliferation, apoptosis, and gene expression (*Aguilar-Solis et al., 2017*; *Liu et al., 2019*). During PVR, binding of TGF-β to its receptor activates PI3K, resulting in the phosphorylation of Akt; activated Akt inhibits glycogen synthase kinase 3β (GSK-3β), promoting EMT in RPE cells (*Shukal et al., 2020*; *Zhang et al., 2018a*). Researchers have found that inhibition or knockdown of GSK-3β promotes cell migration and collagen contraction in ARPE-19 cells, while GSK-3β overexpression and PI3K/Akt inhibitor reverse these cellular responses (*Huang et al., 2017*). Some studies have shown that thrombin can activate PI3K, resulting in increased cyclin D1 expression and RPE cell proliferation, processes that are involved in the development of PVR through PDK1/Akt and PKCζ/mTORC signaling (Fig. 3) (*Lee-Rivera et al., 2015*; *Palma-Nicolás & López-Colomé, 2013*; *Parrales et al., 2013*).

In addition to the PI3K-AKT pathway, other kinase pathways contribute to EMT in cooperation with the SMAD-dependent signaling pathways. In human RPE cells, TGF-β activates TGF-β-activated kinase 1 (TAK1), which subsequently transduces signals to several downstream effectors, including p38 (*Heffer et al., 2019*), JNK (*Kimura et al., 2015*)
and nuclear factor-κB (NF-κB) (*Chen et al., 2016b*), which participate in EMT. *Dvashi et al. (2015)* found that TAK1 inhibitor caused a reduction in both p38 and SMAD2/3 activity, attenuating cell migration, cell contractility and α-SMA expression in TGF-β1-induced RPE cells. Moreover, the ERK MAPK pathway plays a role in TGF-β-induced EMT and cooperates with other signaling pathways in the regulation of EMT in RPE cells. Recent studies (*Chen et al., 2014b*; *Tan et al., 2017*; *Xiao et al., 2014*) have shown that blocking the ERK1/2 pathway inhibits the phosphorylation of SMAD2 and the Jagged/Notch pathway. Inhibition of the Jagged/Notch signaling pathway can alleviate TGF-β2-induced EMT by regulating the expression of Snail, Slug and Zeb1 (Fig. 3); this also suppresses the ERK1/2 signaling (*Chen et al., 2014b*).

The contribution of growth factors other than TGF-β, such as HGF, fibroblast growth factor (FGF), epidermal growth factor (EGF) and platelet derived growth factor (PDGF) should also be factored in with regard to the induction of RPE EMT. These factors bind to and stimulate the autophosphorylation of transmembrane receptors on Tyr, subsequently participating in RPE cell EMT via PI3K/Akt pathway, ERK MAPK pathway, p38 MAPK pathway (Fig. 3) (*Chen et al., 2016a*; *Ozal et al., 2020*). *Chen et al. (2012)* explored the role of Wnt/β-catenin signaling in PVR, and found that when EGTA disrupted contact inhibition in RPE cells, EGF+FGF2 could activate Wnt signaling and increase nuclear levels of β-catenin, which interacts with TCF and/or LEF, leading to cell proliferation (Fig. 3); and EGF+FGF2 cooperated with TGF-β1 to induce EMT through SMAD/Zeb1/2 signaling. Acting together, various inductive signals received by RPE cells from their niche can trigger the activation of EMT programs by individual intracellular cascades or the crosstalk of multiple intracellular signaling pathways.

### Interventions of RPE EMT

Therapeutic interventions against RPE EMT have largely been explored in mechanistic experiments using in vitro cell culture and in vivo animal models. To date, some promising drug candidates have been trialed in preclinical studies of PVR, including TGF-β receptor inhibitors, peroxisome proliferator-activated receptor (PPAR)-γ agonists, retinoic acid receptor-γ (RAR-γ) agonists and methotrexate (*Shu, Butcher & Saint-Geniez, 2020*; *Zhou et al., 2020*).*Nassar et al. (2014)* found that TGF-β receptor 1 inhibitor LY-364947 (LY) attenuates RPE cell transdifferentiation in vitro, and that intravitreal injection of LY completely prevents PVR and TRD in vivo. Evidence is emerging to show that the up-regulation of PPAR-γ expression may be beneficial for the treatment of fibrosis in several organs (*Wang et al., 2019*). *Hatanaka et al. (2012)* reported that PPAR-γ agonist pioglitazone could prevent TGF-β-induced morphological changes and the up-regulation of EMT-related markers in primary monkey RPE cells, through inhibition of the SMAD pathway. Some drugs, including dichloroacetate (DCA) (*Shukal et al., 2020*), salinomycin (SNC) (*Heffer et al., 2019*), resveratrol (*Ishikawa et al., 2015*), protein kinase A inhibitor H89 (*Lyu et al., 2020*) and heavy chain-hyaluronan/pentraxin3 (*He et al., 2017*), reportedly inhibit EMT in an in vitro EMT cell model and prevent PVR development by blocking the activation of theTGF-β pathway. Thus, inhibition of EMT by pharmacological agents may be an effective strategy to prevent PVR development.

## CONCLUSION

Clinical and experimental studies have shown that RPE cells play an important role in PVR. Junctional complexes are crucial for the maintenance of RPE polarity. Under the influence of growth factors and cytokines, RPE cells lose cell–cell contact and apical-basal polarity, and undergo EMT via multiple signaling pathways, which promote cell proliferation, migration, and ECM production. RPE cells further transform into myofibroblasts and form fibrocellular membranes that have contractile activity and strain the retina, leading to tractional retinal detachment in PVR. As a complex refractory blinding disorder, PVR involves multiple signaling pathways and factors. In addition, the specialized polarity of RPE cells is fundamental for retinal homeostasis, and RPE EMT plays a key role in the development of PVR. Nevertheless, further research into the mechanisms underlying RPE polarity and EMT is needed to prevent this devastating complication. A deeper understanding of RPE polarization is fundamental for elucidating the mechanism of EMT initiation and progression, and is essential to exploring the potential pharmacologic prophylactic and therapeutic approaches to PVR. Various factors, such as microenvironmental signals, transcription factors, and epigenetic factors, participate in the regulation of EMT at different molecular levels. Further studies about the detailed molecular mechanisms of EMT are needed to facilitate the development of therapeutic strategies for PVR.

### Funding

This work was supported by the Medical and Health Personnel Special Project of Jilin Province (Grant No. 2019SCZT021) and the Health Service Capacity Improvement Project of Health and Family Planning Commission of Jilin Province (Grant No. 3D5172173429). The funders had no role in study design, data collection and analysis, decision to publish, or preparation of the manuscript.

### Grant Disclosures

The following grant information was disclosed by the authors:
Medical and Health Personnel Special Project of Jilin Province: 2019SCZT021.
Health Service Capacity Improvement Project of Health and Family Planning Commission of Jilin Province: 3D5172173429.

### Competing Interests

The authors declare there are no competing interests.

### Author Contributions

- Hui Zou conceived and designed the experiments, prepared figures and/or tables, authored or reviewed drafts of the paper, and approved the final draft.
- Chenli Shan and Linlin Ma performed the experiments, prepared figures and/or tables, and approved the final draft.

- Jia Liu and Ning Yang analyzed the data, prepared figures and/or tables, and approved the final draft.
- Jinsong Zhao conceived and designed the experiments, authored or reviewed drafts of the paper, and approved the final draft.

## Data Availability

The raw measurements are available in Figs. 1–3.

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
