# Peer review of "Polarity and epithelial-mesenchymal transition of retinal pigment epithelial cells in proliferative vitreoretinopathy"

_PeerJ, doi:10.7717/peerj.10136_

## Round 0.1 · original submission · Major Revisions

Please consider all the issues raised by the referees and include a statement on the method of literature retrieval and selection that should be based on transparent criteria.

Reviewer 1 ·

Basic reporting

This manuscript by Zou and colleagues covers the topic of epithelial mesenchymal transition (EMT) in relation to a ocular fibrotic complication termed proliferative vitreoretinopathy. The main targeted audience is, therefore, people interested in ocular fibrosis. However, as EMT plays a key role in other fibrotic complications as well as in cancer metastasis, the manuscript may have some cross-disciplinary interest.
While there are a fair amount of overlap with past reviews on the topic, the manuscript also describes recent studies that have been published in the last three years.
The Introduction does clearly summarize the subject area, current problems, and the motivation.

Experimental design

The review is organized into subsections but could be better organized into coherent paragraphs in order to avoid jumping around and remove unnecessary repetitions.

Validity of the findings

The authors claims that they have reviewed “emphasizing key insights into potential approaches to prevent PVR” (from the abstract). However, I was not sure which part actually emphasized key insights.
The manuscript conclude with the need for more research on mechanisms involved in PVR, and does not necessarily identify gaps or future directions.

Additional comments

The vast number of cited references clearly demonstrate that the authors covered many articles. While the authors have covered many relevant studies, it becomes a little hard to follow due to the following reasons.
(1) The written English in certain places make it harder to follow. Improvement of the English language will significantly enhance the readability of the manuscript and ensures that the audience clearly understand your text. Improvement in the use of commas, in particular, will be beneficial.
Repeated typos: “RPE cells EMT” should be “RPE cell EMT”
Other typos: extracelluar (line 128), Samd (should be Smad, Fig2)
Suggested change to use of “release (line 121)”, “therefore (line 124)”, “transmit (line 128)”, “through (line 274)”
Consider rewriting sentence: Line 226 (;they change ~)
(2) Description of studies using RPE and other cell types are often intermingled without clarification. Furthermore, studies that has been conducted on the human RPE cell line ARPE-19, which never becomes truly polarized, is often treated as RPE cells within the manuscript. It will be very helpful if the authors distinguish between non-RPE, RPE and ARPE-19.

Section 3. RPE and Fibrocellular Membrane – please clarify between “differentiated RPE” and “de-differentiated RPE (EMT-RPE)” as the “RPE” that contributes to PVR membrane formation is the EMT-RPE.

Minor comments
Not sure if Line155-165 is necessary as it does not seem to have relevance to EMT or PVR.
Line 261-262 – explain the function of cofilin and how it is regulated by phosphorylation.
Line 295-303 reads like an add-on after writing the manuscript. It will be great if it can be incorporated into the section above.

·

Basic reporting

The review is within the scope of the journal.

there are numbers of reviews that are similar to the current review, however, previous review did not make a clear connection of the RPE Polarity to EMT.

The Introduction of the review is adequate.

Experimental design

It is a comprehensive review, it covered most of the information in the field.

The uncovered information, not cited reference and the suggestion for the review organization are listed in the following section of General comments for the author.

Validity of the findings

General speaking, the review is well developed and supported argument that meets the goals set out in the Introduction.

The conclusion did not include the contents of unresolved questions / gaps.

Additional comments

This is an interesting review paper; The authors summarized some of previous publications in the study of the role of RPE EMT in the pathogenesis of PVR. the diagrams look nice.
Following questions should be addressed before the consideration of the paper to be published in the journal.
Major points:
1. Lack of the information of Clinical evidence of RPR EMT in the pathogenesis of PVR
2. the authors mentioned the Polarity of RPE, what are the characteristics of RPE Polarity physiologically and what is the phenotypes changes in the condition of EMT? In factor, the authors put the RPE Polarity in the remarkable position, however, there is not much description of the relevance of RPE Polarity to EMT, in order to make its title consistence with contents, the review should be reconstructed and extended.
3. the role of Epigenetic factors in the contribution of RPE EMT has not been incorporated into the review paper.
4. Review is not only a pile up of previous publications but also your view and comments should be included. In the section of conclusion, the authors should incorporate what are the unsolved question in the field and what is the speculation.

Minor:

1. There are 3 types of EMT, in the current review which type EMT refers to?
2. What are the markers in RPE cells when EMT occurs; in the condition of EMT, which molecular markers are obtained and which lost?
3. Interruption of junction of RPE is a critical step in the induction of EMT, there are some key publications about this; among the previous study, HGF has been shown to play an important role in the interruption of RPE junction, treatment of RPE explants with HGF results in rapid disassembly of tight and adherens junctions associated with loss or redistribution of junctional proteins, decreased TER, and increased migration of RPE cells from the monolayer (Jin et al. IOVS, August 2002, Vol. 43, No. 8,2782-2790; Invest Ophthalmol Vis Sci. 2004;45:323–329). Those informative message has been neglected
4. there are much more transcription factors involved in the induction of EMT, the authors need to look up previous publication closely.
5. line 304, RNA and EMT, it should be miRNA not RNA
6. It may be better to put section 3 “ RPE and Fibrocellular Membrane” in the second place.
7. no information about the intervention of RPE EMT.

Reviewer 3 ·

Basic reporting

There are many reviews about RPE cell EMT and PVR, but previous reviews did not pay much attention to the relationship between RPE polarity and EMT. So I think it’s reasonable for this review to be published in the journal after revision, the review is within the scope of the journal. The introduction does adequately introduce the subject.

Experimental design

This is a review manuscript which content is within the Aims and Scope of the journal. The survey methodology is consistent with a comprehensive, unbiased coverage of the subject. But I think there is logical problem in this review manuscript. And not all sources were adequately cited.

Validity of the findings

The review is a well developed and supported argument that meets the goals set out in the Introduction. But the conclusion doesn’t include the contents of unresolved questions / gaps/ future directions.

Additional comments

To be frank, this is an interesting review manuscript. Although there are many reviews about RPE cell EMT and PVR, but previous reviews did not pay much attention to the relationship between RPE polarity and EMT. But there still has some questions, the following questions should be addressed before the manuscript to be fully considered to be published in the journal.

Major points:
1. The title include the polarity of RPE cell, it’s the main difference from previous review about EMT and PVR, but what is polarity of RPE, and what’s the change once EMT occurs, what’s the real relationship between the polarity of RPE cell and EMT in PVR, all this questions were not answered clearly. The author should adding a new part about RPE polarity and EMT.
2. The manuscript mentioned the relationship between lncRNA and EMT, is it necessary in this manuscript, if it’s necessary, why not adding other epigenetic factors in this manuscript?
3. In the conclusion, the authors should mention what are the unsolved questions in the field and what is the future direction.

Minor points:
1. There are some new papers about EMT and RPE cells, which also summarized the EMT of RPE cells and transcription factors of EMT, the authors should find new reference from recent publication.
2. The logical structure should be adjusted, I think it would be better to change the place of part2 (RPE and EMT) and part 3 (RPE and Fibrocellular Membrane).
3. As we all known that there are 3 types of EMT, the authors should mention that which kind of EMT happens in PVR in this manuscript.
4. The figures looks good, but there are some spelling mistakes, such as in Figure 2, Samd2/3 should be replaced with Smad2/3, Samd4 should be replaced with Smad4.

---

## Round 0.2 · Minor Revisions

The first revision has adequately answered most of the issues raised. Please address the final outstanding issue from Reviewer 2.

·

Basic reporting

no comment

Experimental design

no comment

Validity of the findings

no comment

Additional comments

The authors answered all questions raised by reviewers, the manuscript was improved a lot.
however, There is one concept which need to be corrected, in line 296, the authors mentioned “Unlike transcription factors, epigenetic modifications are more stable/long-term”. the statement is not right;in fact, Epigenetic modifications are highly dynamic process in developmental biology and pathogenesis of diseases.

Reviewer 3 ·

Basic reporting

The authors have made good revision, no further concern.

Experimental design

The authors have made good revision, no further concern.

Validity of the findings

The authors have made good revision, no further concern.

Additional comments

The authors have made good revision, no further concern.

---

## Round 0.3 · accepted · Accept

You have adequately addressed the issues raised by the referees.